# GIS-Based Multi-Objective Routing Approach for Street-Based Sporting-Event Routing

Young-Joon Yoon [1], Seo-Yeon Kim [1], Yun-Ku Lee [1], Namhyuk Ham [2,*], Ju-Hyung Kim [1] and Jae-Jun Kim [1]

1    Department of Architectural Engineering, Hanyang University, Seoul 04763, Republic of Korea; kim1723s@gmail.com (S.-Y.K.); hook1996@hanyang.ac.kr (Y.-K.L.); kcr97jhk@hanyang.ac.kr (J.-H.K.); jjkim@hanyang.ac.kr (J.-J.K.)

2    Department of Digital Architecture and Urban Engineering, Hanyang Cyber University, Seoul 04763, Republic of Korea

*    Correspondence: nhham@hycu.ac.kr

**Abstract:** This study proposes a decision-making framework that integrates a routing model based on the geographic information system (GIS) and a genetic algorithm into a building-information modeling (BIM) environment to overcoming the limitations of the planning process of traditional street-based sporting events. There is a lack of research on improving the manually conducted decision-making processes for street-based sporting events. Moreover, previous routing studies were limited to GIS environments, and proposals for decision-making models integrated with BIM environments are lacking. In this study, the applicability of the framework was verified by presenting the variables of the existing GIS-based routing model as environmental variables to consider the impact of street-based sports events on a city. The evaluation model for the route selection was parameterized independently, such that its priority could be changed according to the user's needs. Moreover, we integrated the data into BIM to create and analyze models that assess urban effects. This method is a decision-making system for policymakers and event planners to promptly conduct initial venue surveys through the technological integration of GIS–routing–BIM. Additionally, the GIS stipulated in this study can be applied to other cities. The Gwanghwamun area of Seoul, South Korea, was selected as the case study.

**Keywords:** GIS; routing; multi-objective problem; street-based event; GIS-to-BIM; optimization





## 1. Introduction

### 1.1. Research Background

Events staged in urban public spaces effectively create a sense of social solidarity by inducing citizens to participate [1]. Moreover, events staged in familiar spaces provide new experiences for citizens [2], and they can be used by policymakers as media for effective message delivery [3]. Smith [4] asserted that mega events such, as sporting events, can be used to effectively deliver urban messages by fostering citizens' voluntary participation.

However, mega events in public spaces can restrict, control, and damage the host's space, thereby eroding a city's public life [5]. Street-based sporting events, such as cycling events and marathons, inevitably have a larger impact, since they utilize the host city's roads as venues [1]. Street-based events generally require roads to be closed for longer periods than the number of days they actually last. Restrictions on public spaces are inconvenient for citizens, although this problem can be solved by environmental design and by considering the impact of events on city centers.

The Formula E World Championship is a sustainable urban race that aims to develop and promote low-carbon, low-emissions technology for electric vehicles [6]. The event first took place in 2014, and 99 races have been held over the last eight years in major host cities around the world [6]. The event occurs annually, and the hosting trend has continuously increased, except for the 2019/2020 seasons, which were affected by the

COVID-19 pandemic. Indeed, the event, which started with 11 rounds in 10 cities in its first season, grew to 16 rounds in 11 cities by 2022. These data indicate the possibility of hosting more street-based sporting events in the future.

The event offers participants and visitors a dynamic and exciting experience through urban racing. However, compared to street-based sports, such as cycling events and marathons, significant construction is involved, and the use of public roads is restricted for long periods. For example, traffic islands, flower beds, and speed bumps that interfere with the host city's roads must be removed. Indeed, when the championship was held in Jamsil, Seoul, in 2022, the use of public roads was restricted for 47 days for a 4-day race. In sum, because the championship has a direct and critical impact on host cities, environmental design based on the analysis of a given city is a key requirement in the event-planning stage. However, the route selection for existing street-based sports events is based on the subjective judgment of policy makers rather than objective data, and there is a tendency to value the success of the event rather than its impact on the city.

The Fédération Internationale de l'Automobile (FIA) provides guidelines, including regulations on track design for events, through event specifications. However, problems with street-based sporting events, such as the Formula E World Championship, include the selection method for the initial venue and the limited decision-making process. Venues and track routes are determined by the local government of the host city. Local governments lack motorsport expertise and have difficulties in selecting reasonable routes for races. Moreover, feasibility reviews of selected venues and track routes are conducted under limited conditions. As a result, considerable time is consumed in making decisions regarding the hosting of these events, and it is impossible to analyze various alternatives, resulting in monotonous track composition. Additionally, design changes may occur because information is not reviewed in advance, and routes that do not fit the host city may be designed. These obstacles can reduce the positive impact of events and increase inconvenience for citizens.

For example, in the 2022 Jamsil E-Prix, it took four months to explore the first track route because of the manually conducted event-planning process, and the initially selected track route was changed just before the event due to insufficient evaluation of the impact on the downtown area. As a result, design changes were made throughout the track and competition overlays, and additional costs were incurred owing to renegotiations with the government, re-approval by the FIA, and urgent construction for the competition schedule. Figure 1 shows the initial and final track layouts. The red box is a section that has been deleted by the public road impact assessment, and the track inside the stadium has been lengthened owing to the deletion of that section. In other words, for street-based sports events, such as the Formula E World Championship, because of the nature of events that are held for a relatively short period of time, the evaluation of the impact on the downtown area is overlooked, and the manually conducted planning process requires significant time for route search and review and, sometimes, Elements that are not considered can cause errors. Moreover, research on digital-based processes for street-based sporting events has not been conducted on current routing and GIS fields. Therefore, a digital process that can route street-based events based on city data is needed. This should be accompanied by an analysis of how planned routes affect host cities.

The data on cities required for the planning of street-based sporting events are provided extensively by geographic information systems (GIS). Based on these data, a race route can be searched by using an optimal-path-search algorithm. Building-information modeling (BIM) also supports decision making in the early stages of a project through clash reviews, quantity takeoffs, and visualization. Moreover, the generated BIM data can be utilized throughout the project's life cycle. In previous studies, the dyadic technological combination of GIS–routing/GIS–BIM was actively dealt with. However, there is a lack of research on decision making based on the technological integration of GIS–routing–BIM. In addition, previous GIS-based routing research focused on routing, and research on

BIM-environment integration to explore the impact of derived routes on cities is lacking. Therefore, it is necessary to develop a decision-making model.

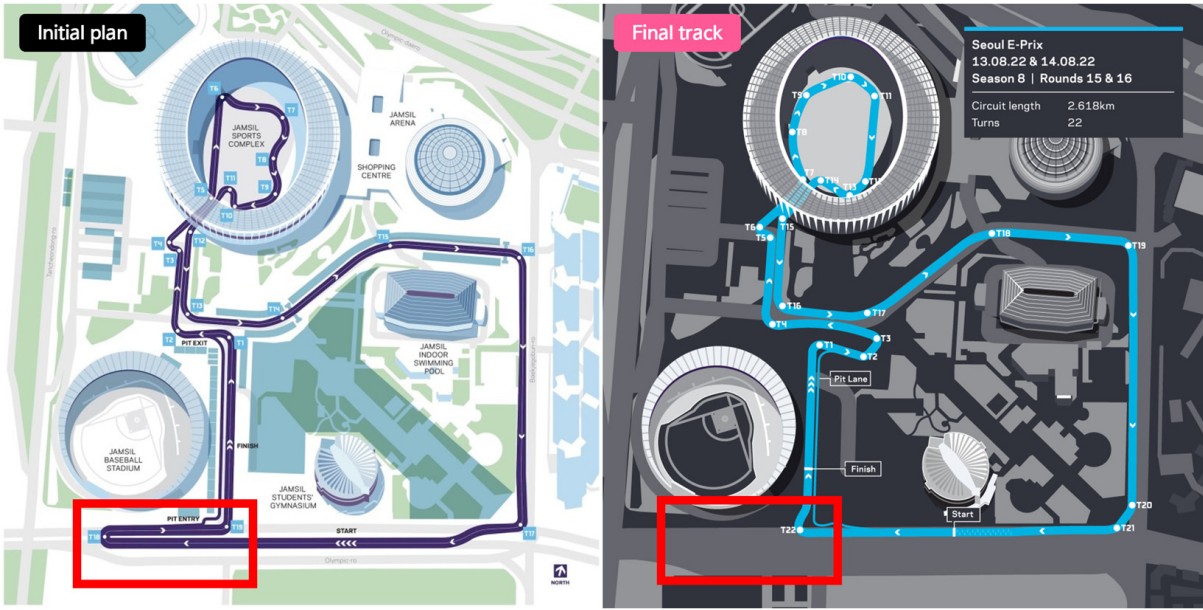

**Figure 1.** The initial and final track layouts of 2022 Jamsil E-Prix.

The aim of the present study is to develop a data- and algorithm-based routing process for street-based events and to propose a framework to evaluate their impact on a given host city. We created a database of host cities in GIS and developed an algorithm to search for optimal event routes with a low impact on the city. Moreover, we integrated the data into a building-information model (BIM) to create and analyze models that assess urban effects. We proposed the present process to facilitate rapid decision making during the planning phases of street-based events. Additionally, if the process conforms to the GIS stipulated in this study, it can be applied to other cities.

This paper includes four subsequent sections. In Section 1, we define the route-selection factor as a parameter and explain how we constructed the GIS-based optimal-routing algorithm. In Section 2, we describe the techniques used to collect and process GIS datasets and a framework for developing algorithms. We clarify the approaches to solving the multi-objective optimization problem in this section. In Section 3, we use the case study to show the results of applying the proposed process, with a focus on a specific region. We selected the Gwanghwamun area of Seoul as a case study. In Section 4, we examine the study's implications. Section 5 summarizes the study's findings and limitations and addresses future research.

### 1.2. Literature Review

### 1.2.1. Limitations of the Street-Based Event-Planning Process

The planning of a typical street-based sporting event is under the auspices of competent government and event organizers. Governments lack expertise in sports, and event planners focus on the success of events [5]. The difference between these two perspectives creates a significant amount of time in decision making because of the differences in requirements. Moreover, when an event planner lacks urban and social information about a venue, it causes problems in devising an event route that is unsuitable for the conditions of the host city [1]. In addition, since street-based sporting events require the occupation of public roads for a relatively short period of time, the negative impact is sometimes not considered in the success of the event, even if citizens are inconvenienced.

Route searches for existing street-based sports events are subjective and inefficiently planned through media, such as web maps. First, a route suitable for the physical conditions

of the track required for a sports event was searched through a web map. Subsequently, surveys on the current status of the route explored through on-site inspections and the investigation of obstructions were conducted. Field visits are time-consuming. In fact, four weeks were spent on field research at the 2022 Jamsil E-PRIX. In addition, if factors that make it impossible to hold an event are identified during site surveys, the route is searched again. Such manual route searches in the planning stage are inefficient and lack objective evaluations of the selected route, and searches for various alternatives are impossible. Street-based sporting events, such as bicycle races and marathons, as well as the Formula E World Championship, are very diverse and frequent all over the world. In other words, it is necessary to propose a digital-based objective decision-making process to solve inefficient processes during the planning stages of numerous street-based sports events.

The route searches for current distance-based sports event can be performed more quickly and efficiently using the GIS-based routing methodology suggested in various previous studies. The GIS environment provided by the government includes information regarding the infrastructure of a city. This can be used to efficiently conduct obstruction investigations in a digital environment during the early stages of planning. Moreover, it is possible to search for a route that considers the city infrastructure in the initial route search. In addition, through the integration of GIS and BIM environments, a city model can be visualized, and an evaluation model can be created [7], which can be used as a decision-making model suitable for street-based sports events. Therefore, in this study, we propose a comprehensive decision-making framework based on GIS–routing–BIM, which can be used to solve the problems of current street-based sports events; to this end, we selected a specific city, conducted an experiment, and discuss the results in this paper.

### 1.2.2. The Vehicle-Routing Problem with GIS

In the literature on the vehicle-routing problem (VRP), GIS is seen as a powerful database. The GIS is a platform used to store, manage, and visualize large amounts of city-related data and to support analyses by exploring relationships between various pieces of information [8–10]. Some countries offer large amounts of open-source data on specific geographic details, road infrastructure, and transport [11].

Against this background, GIS data have been utilized in the literature related to pathfinding and path evaluation for the past several decades [12]. The VRP field for route optimization is usually based on GIS-based network analysis [13]. In addition, routing research using GIS has been conducted in the railway, waste, distribution, and transportation sectors. Al-Hammedawi et al. [14] examined optimal railway tracks, and Kanchanabhan et al. [15] proposed the optimization of a solid-waste-collection route.

The simplest route-selection model is based on the shortest path [16]. However, minimizing the general cost, time, or distance might not be the only influence governing route selection [17].

An important factor in the route-selection model is the purpose of the system. Therefore, it is necessary to define selection criteria to determine a suitable route for a given purpose [18].

The aim of a street-based sporting event in a city is the dynamic presentation of the event based on the safety of citizens. The routing of the event involves several factors. Algorithms that support the general shortest–longest distance or minimum–maximum action are not available. In sum, an algorithm that satisfies or converges toward a multi-objective factor should be used.

Evolutionary (e.g., genetic) algorithms should be used to determine the optimal path that solves the multi-objective problem (MOP) [19]. The literature mentions the fusion methodology of the Dijkstra algorithm and the evolutionary algorithm. In particular, the non-dominated sorting algorithm-II (NSGA-II) [20], used for path multi-objective optimization based on non-domination levels, presented by Srivastava et al. [21], is an effective technique for solving the MOP. The simulation results for various factors using NSGA-II show that it is superior to other optimization approaches [22].

Recently, the NSGA-III methodology for large-scale data optimization was proposed. This methodology is used to overcome high data-calculation costs, which are created by the current NSGA-II approach [23]. For example, Xue et al. [24] proposed the NSGA-III methodology for optimizing biomedical ontology alignments, in which concepts of similarity measures cannot be independently distinguished under any circumstances, such as biomedical information systems.

However, the NSGA-III approach is generally used for many-objective optimization (MaOP) with more than four objectives [25], and computational experiments in MOP with fewer than ten showed that NSGA-III outperformed NSGA-II [26]. In accordance with these results, the NSGA-II approach was adopted in this study.

Moreover, since the variables considered in GIS data vary according to the purpose, previous studies cannot be directly used for distance-based event-path searches.

In other words, in this study, the NSGA-II methodology presented in previous studies was applied to select the optimal route that can satisfy the elements required by street events and minimize the impact on cities. Furthermore, we propose a comprehensive decision-making process, including the quantitative evaluation of the derived routes by linking them with BIM.

### 1.2.3. Route-Selection Factors: Route Variables and Environmental Variables

For the routing in this study, we used route and environmental variables. We fed the route variables into a modified Floyd–Warshall algorithm to generate multiple routes. We entered the environmental variables into NSGA-II's generic crossover and mutation operators to produce offspring. The NSGA-II applies the genetic algorithm (GA) to create new offspring to find optimized solutions between route and environmental variables [27].

A route variable is the physical condition of a road required for a vehicle's progression at a sporting event. The variables that we used consisted of the road's width, the number of turns, the turning radius required of the vehicle, and the total length of the track. The physical conditions of the Formula E racetrack and the selection factors for the route can be confirmed through FIA's event specifications and interviews with professional track designers; the FIA has a relatively flexible index that responds to different environmental contexts for each host city. In this study, we defined the index as a quantitative parameter for implementing the optimization algorithm.

Environmental variables minimize an event's impact on a city and help to reduce inconvenience for citizens by identifying facilities that are blocked due to the event and setting weights according to their impact on citizens' lives. For example, planning committees should avoid holding events on roads adjacent to educational facilities, such as kindergartens and elementary schools. In addition, the roads used should not block access for vehicles from adjacent disaster-prevention facilities, such as police and fire departments. In this way, we listed the facilities necessary for citizens' daily lives and set the avoidance weight for each facility. We directly fed this information into the optimal criterion for the path extracted through the algorithm.

Table 1 defines the race-route and environmental variables. The index is composed of three levels based on the order of the algorithm development and data processing. The data consisted of parameters and values.

The data in Level 1 encompass parameters for selectively extracting roads drawn from the city's road network that can be used in the routes for sporting events. The Level 2 data involved parameter inputs for the path-search algorithm. The data in Level 3 included environmental variables that are inputs for routes optimization. We defined the environmental variables as a list of facilities to be avoided to prevent inconvenience for citizens. Each indicator had a weight set to reach the optimal value according to priority and was parameterized so that the user can enter the desired value. Section 3 outlines the definitions of the weights set.

**Table 1.** The definitions of the variables.

| Category | No. | Parameter | Definition | Value | Unit |
|---|---|---|---|---|---|
| Level 01: Route variables-I | 1 | Width of the road (min.) | - State of use as track minimum width of road (excluding pedestrian roads)<br>- Occupancy width, including temporary facilities, such as safety fences and pedestrian detours | 12 | M |
| | 2 | Pass through intersections | - Whether route through intersections is possible | Y | - |
| | 3 | Pass through bridges | - Whether route through bridge is possible | N | - |
| | 4 | Pass through tunnel | - Whether route through tunnel is possible | N | - |
| | 5 | Height obstacle | - Minimum height of height obstacle (e.g., bridges, other structure, etc.) above the routes | 4.5 | M |
| Level 02: Route variables-II | 1-1 | Track length | - Total length of one lap of the event track | 2.4 | KM |
| | 1-2 | Tolerance range of track length | - Tolerance range of one lap of the event track | $\pm0.3$ | KM |
| | 2-1 | Number of turns (min.) | - Minimum number of turns in one lap | 14 | Turns |
| | 2-2 | Number of turns (max.) | - Maximum number of turns in one lap | 22 | Turns |
| | 3 | Radius of turning (min.) | - Minimum turning radius for racing vehicles (outer path) | 12 | M |
| | 4 | Radius of turning (max.) | - Radius identified as a turn | 120 | M |
| | 5 | Buffer-interval length (min.) | - Minimum length of start grid before start line<br>- Minimum length of buffer section for stopping after passing the finish line | 180 | M |
| Level 03: Environmental variables | 1 | Educational institutions | - School (kindergarten, elementary school, middle school, high school, university, etc.), education center, vocational training center, academy, laboratory, library | | |
| | 2 | Child- and geriatric-welfare institutions | - Child-related facilities, elderly-care facility | | |
| | 3 | Public institutions | - government building, provincial government building, city hall, police department, fire department, post office, facility for public affairs | | |
| | 4 | Medical facilities | - Hospital, quarantine facility | | |
| | 5 | Transportation facilities | - Passenger terminal, cargo terminal, railway station, airport facility, port facility | | |

For this study, we configured the city where the event is to be held and the city's central point (the main bases equipped with all the infrastructure for the event) for the user to select.

### 1.2.4. BIM on GIS

The BIM environment is expressed in digital data by integrating a building or city's shape and property information; it is used for planning, design, construction, and management throughout a project's lifecycle [28] A major benefit of the BIM environment is the fact that it supports decision making by project stakeholders [29].

The BIM environment includes a building's information, and GIS supports powerful analyses by providing geospatial data about cities [30]. Accordingly, the integration of GIS and the BIM environment is under study to implement a digital-twin-based smart city [31]. For example, the spatial-information sector uses GIS and the BIM environment to assess traffic noise and building-flood damage, to visualize geographic information in a 3D manner, and to simulate floods [32–34]. In addition, the architecture, engineering, and construction industries integrate and utilize the BIM environment to manage emergency

responses and construction supply chains, as well as to plan highways [35–37]. In this way, through BIM–GIS integration, various analyses and decision-making processes can be supported from a city-level perspective.

The approach of BIM–GIS integration mainly relies on data exchange between the two systems. While the GIS approach is used to code environmental information based on geographical references, the BIM approach is used to code buildings in terms of their components to create parametric descriptions [38]. In terms of information flow, since the GIS conversion of BIM data causes geometric information loss, a conversion tool that minimizes this loss is needed [39]. The Open Geospatial Consortium (OGC) and buildingSMART International (bSI) analyzed IFC, CityGML, and LandInfra to establish a data standard for the integration of GIS and BIM and presented a built-environment data-integration framework [40].

However, according to Zhu and Wu [41], in a study of BIM/GIS data integration from the information-flow perspective, BIM and GIS can be integrated at the raw data level and, compared to BIM-to-GIS data conversion, GIS-to-BIM is relatively free from problems related to data conversion.

In this study, we integrated GIS and the BIM environment to examine their impact on cities. We established data conversion from GIS to the BIM in the BIM environment by extracting GIS shape files into computer-aided design (CAD) and Excel data. This allowed us to integrate GIS datasets for the elements and routes that constitute the city studied into the BIM environment. We used an integrated BIM environment to assess the impact on urban areas.

## 2. Methodology

In this section, we propose a street-based sporting-event-routing process using GIS data and algorithms to prepare for potential street-based events in cities. Based on open GIS, we generated information on the host city, searched for a route using a modified version of the Floyd–Warshall algorithm, and derived an optimal route through the NSGA-II approach. In addition, we evaluated the impact on the city through an automated process that incorporated the derived dataset into the BIM environment.

Figure 2 illustrates the process of the proposed routing model; it consisted of the following: (1) data collection and processing; (2) route exploration and optimization; and (3) city-impact assessment through BIM-model creation.

We collected and processed the data in a GIS-software (Q GIS version 3.26.0) environment. Based on the selected venues, we generated a GIS dataset for transportation networks, buildings, and facilities within a specific range. We selectively extracted the generated data based on information system. We then pre-processed the extracted data so we could reduce the amount of computation by lowering the amount of data through pre-processing. Next, we explored a route based on the route variables and derived an optimal route grounded in the weights of the environmental variables. We integrated the derived routes and elements constituting the city into the BIM environment through the GIS-to-BIM process. We evaluated the impact on the city based on the relationship between the urban infrastructure and potential event routes. The impact assessment determines whether there is interference between the urban infrastructure and routes, and collects and extracts the quantity of interfering elements; it also supports stakeholder decision-making by visualizing data.

### 2.1. Data Collection and Pre-Processing Approaches

The collection of the data included all the GIS data about the city, including the transportation network, to evaluate the impact of optimized event routes on the city. We obtained the data from the National Geographic Information Institute (NGII), which manages spatial information under the leadership of the South Korean government. The NGII provides city data in the following categories: traffic, buildings, facilities, vegetation, water systems, topography, boundaries, and cycles.

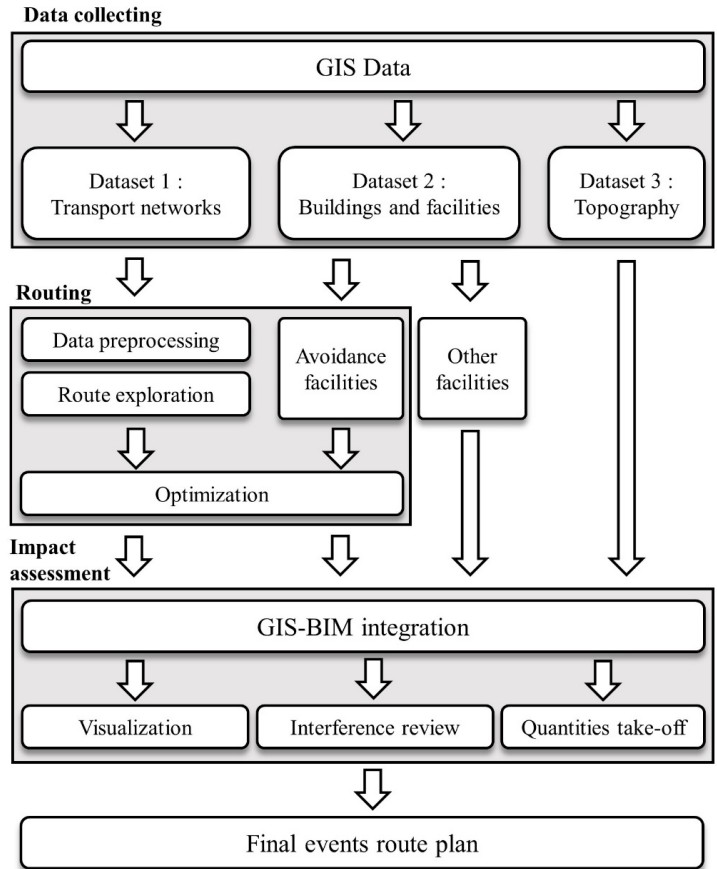

**Figure 2.** Framework of the proposed routing model.

Table 2 shows the built data categories and the detailed data composition. We required GIS data for four different elements: (a) transportation networks; (b) buildings; (c) facilities; and (d) topography.

**Table 2.** Data categories and composition.

| Category | Dataset | Data Type |
|---|---|---|
| Transportation networks | Roads | Line shapefile |
| | Pedestrian roads | Line shapefile |
| | Attribute information | Road ID, road width, traffic type, number of lanes, existence of a median strip, materials of road pavement, classification of road |
| Buildings and facilities | Educational institutions | Polygon shapefile |
| | Child- and geriatric-welfare institutions | Polygon shapefile |
| | Public institutions | Polygon shapefile |
| | Medical facilities | Polygon shapefile |
| | Transportation facilities | Polygon shapefile |
| | Attribute information | Building (or facility) name, usage, floors, height |
| Topography | Contour line | Line shapefile |

(a) We generated transportation-network data centering on road routes composed of lines and points where road routes intersected. Each route contained attribute information for effective road management (e.g., road ID, road width, traffic type, the number of lanes,

and the existence of a median trip); this can usually be interpreted as the physical structure of the road, and the physical relationship between lines can be extracted (e.g., the length of lines and the angles between them). We used traffic data to extract roads that can be used as tracks and as input parameters for routing algorithm.

(b) We used building and (c) facility data to configure the environmental variables. The shape can be checked through the outlines of the elements in the GIS data, which can be visualized through the number of floors and height information. Furthermore, road routes to be avoided (e.g., educational and medical facilities) can be identified based on the usage information of the building.

(d) Topographic data are usually composed of contour lines, which constrain the permitted slope of the route based on the terrain-slope information; they also support stakeholder decision making by visualizing the topographic data.

We set the range of data collection to a radius of 2 km from the main point of the event, since the Formula E racetrack is a closed route and the maximum length of one lap of the competition is 3.4 km. This is the scope for reducing the amount of data and limiting computer operations.

The user selects the central location of the event. In event planning, sometimes, there is a need to designate an iconic building or space within a city as the main focal point for dramatic production. In addition, temporary facilities, such as TV compounds, pit garages, and medical centers should be located at the base of the event. Hence, we randomly selected the base of the event according to the user's needs and collected data rooted in the base.

We integrated the data into a network dataset using the open-source software program, QGIS. Subsequently, depending on how the data were to be employed, they were classified for use in optimal routing or for evaluation and visualization. We extracted the classified data as CAD and Excel files. In general, CAD files contain shape information and Excel files contain property information. The routing process processes GIS data in the Rhinoceros 3D software (version 6) environment to effectively perform route discovery based on the Floyd–Warshall algorithm, as well as routing optimization through the NSGA-II approach.

## 2.2. Evaluation Model of Route-Selection Factors

Evaluation factors are important in the MOP-based routing model. We built an evaluation model for route selection based on the event specifications provided by FIA and expert interviews. However, the selection of avoidance-model factors to minimize the impact on cities is very broad and depends on the decision-maker. Accordingly, we parameterized the evaluation model for route selection such that the priority of the evaluation model could be changed according to the user's needs.

Table 3 lists the priorities of each parameter applied to the routing process. In MOP approaches, weighting is commonly used to integrate multiple objectives [42]. Chang [43] utilized a weighted method to integrate multiple goals, and Kaewfak et al. [44] combined 0–1 goal programming with an analytic hierarchy process to determine weights to generate optimal routes. In this study, we weighted the evaluation model according to the priority factors. Indicators that needed to meet certain conditions were assigned a value of 0 or 1, and indicators with flexibility were classified according to their importance into percentiles within the range of [0 to 1].

In addition, we set at least one road adjacent to avoidance facilities (e.g., educational institutions and medical facilities) as impossible to select according to the environmental variables. Furthermore, among the urban infrastructure located on the ground, we defined areas frequently used by pedestrians through a literature review [45]. Thus, facilities that affected walkability due to the event were the main targets for monitoring, and a high weight was given to the impact assessment [46,47].

**Table 3.** Priorities for routing.

| Category | Weight | Parameter | Value | Unit |
|---|---|---|---|---|
| **Level 01**: Route variables—I | 1.0 | Width of the road (min.) | 12 | M |
| | 1.0 | Pass through intersections | Y | - |
| | 0.0 | Pass through bridges | No | - |
| | 0.0 | Pass through tunnel | No | - |
| | 1.0 | Height obstacle (min.) | 4.5 | M |
| **Level 02**: Route variables—II | 0.5 | Track length | 2.4 | KM |
| | 0.6 | Tolerance range of track length | $\pm0.3$ | KM |
| | 0.3 | Number of turns (min.) | 14 | Turns |
| | 0.3 | Number of turns (max.) | 22 | Turns |
| | 0.8 | Radius of turning (min.) | 12 | M |
| | 0.7 | Radius of turning (max.) | 120 | M |
| | 0.7 | Buffer interval length (min.) | 180 | M |
| **Level 03**: Environmental variables | 0.9 | Educational-institution avoidance | | |
| | 0.8 | Child- and geriatric-welfare-institution avoidance | | |
| | 0.8 | Public-institution avoidance | | |
| | 0.9 | Medical-facility avoidance | | |
| | 0.9 | Transportation-facility avoidance | | |

*2.3. Routing Framework*

We explored the optimal route of the event track using road-network data built using GIS software. We wrote the path-search algorithm in the C# language in the Grasshopper environment, which is an application-programming interface (API) of Rhinoceros 3D.

The framework of the routing and optimization processes according to the order of data processing was as follows:

1. If the width of the road was less than or equal to the required width of the track, we deleted the data from the route list.
2. If the classification of the road route was bridge or tunnel, we deleted the data from the route list.
3. If there was only one road adjacent to the avoidance facility defined in Section 3.2, we deleted the data from the route list.
4. We deleted completely discontinuous and dead-end roads that did not meet the criteria from the route list. Section 4 outlines the definitions of fully discontinuous and dead-end roads where the criteria were not met.
5. The user inputs a node as the primary base. The event track is a closed path. Hence, a path that starts from the node and returns to it is searched. A routing algorithm randomly explores the path between all pairs of vertices in an edge-weighted directed graph and builds a regression path. If the width of the road exceeded twice the required width of the track, it was recognized as a turnaround route. However, it was not selected as a path if the turn radius of the turn node was not satisfied.
6. We entered all studied regression routes into the GA process. The GAs apply genetic operations, such as population representation, selection, crossover, and mutation, to find optimal results for complex problems [48]. In order to define the optimal outcome, we evaluated the route by entering the weight for each parameter stated in Section 3.2. We then subjected the evaluated routes to genetic operations to create new offspring and find more optimized solutions.
7. The derived optimal-route list was returned in the form of a closed curve.

The following are the main formulas of path-search used in this study, and Table 4 presents the symbols used in the formula and their explanations.

**Table 4.** Symbols and their explanations.

| Symbol | Explanation |
| --- | --- |
| ra | ath path |
| la | ath path length |
| ta | ath rotation node |
| Ol | Objective length |
| Ot | Objective number of turn |
| Otn | Objective buffer (last path) length |
| B | Buildings interfering with the path |
| Be | Educational-institution avoidance |
| Bc | Child- and geriatric-welfare institutions |
| Bp | Public institutions |
| Bm | Medical facilities |
| Bt | Transportation facilities |
| Wn | Weight |
| 0.9 | Medical facilities |
| 0.9 | Transportation facilities |

$f1$ (Route variables—I) [24]: $f1$ determines whether the condition is met. If the condition is not met, the route is deleted, and if the condition is met, the route is output as an available route.

$f2$ (Route variables—II):

$$\text{Minimize } f2 = \frac{O_l - |\sum_{i=1}^{n} l_i|}{1 - W_l} + \frac{O_t - |\sum_{i=1}^{n} t_i|}{1 - W_t} + \frac{\ln - O_l}{1 - W_l} \tag{1}$$

$f3$ (Environmental variables):

$$\text{Minimize } f3 = \frac{\sum_{i=1}^{n} B_e i}{1 - Wn_e} + \frac{\sum_{i=1}^{n} B_c i}{1 - Wn_c} + \frac{\sum_{i=1}^{n} B_p i}{1 - Wn_p} + \frac{\sum_{i=1}^{n} B_m i}{1 - Wn_m} + \frac{\sum_{i=1}^{n} B_t i}{1 - Wn_t} \tag{2}$$

A random closed path is generated based on the available paths derived through $f1$. Next, based on the path explored through $f2$ and $f3$, a weighted score is derived. Once scored, a pathway is the first offspring, and genetic algorithms generate other offspring to find the minimum score. At this time, the parameters are objective length, objective rotation number, objective buffer (last path) length, and weight.

Thus, the optimal route reflecting the path variable and environment variable is derived as follows:

$$\min f = \{f2, f3\} \tag{3}$$

### 2.4. GIS–BIM Integration and Impact Assessment

We performed visualization, interference review, and quantity take-off by constructing the city's GIS data and optimal route in the BIM environment, which can effectively determine the impact of street-based sporting events on cities. In this process, it is possible to collectively check road infrastructure that needs to be removed or relocated owing to interference with the route.

Urban infrastructure built in GIS were classified into shape and attribute information. We built the shape in the BIM environment from the CAD file, extracted the attribute information into Excel, and entered it according to the BIM information-classification system. In this process, BIM software involved Revit 2023, and BIM-model generation according to the information-classification system was automated through Dynamo's Revit API.

The BIM models were classified as unit and master projects. Unit projects were built independently by dividing them into (a) urban infrastructure and (b) event routes. We linked the built-unit project to the master project. In a master project, the relationship between linked unit projects can be determined and employed for interference review, quantity take-off, and visualization.

In this process, we added a function to automate the barriers and fences employed in the event by extracting the parameters of the path for visualization. In general, because certain objects were continuously placed in the protective wall or protective fence, a unit object was created in the BIM model and configured to be created based on the path.

## 3. Results

### 3.1. Case Study: Project Description

We selected the Gwanghwamun area, located in Jongno-gu, Seoul, as the case study. The Seoul authorities have explored Nodeul Island and Gwanghwamun as candidates for the next venue. Nodeul Island is an artificial island located on the Han River; access to the public is restricted, so there is no road infrastructure. In contrast, the Gwanghwamun area is a typical modern urban zone with a wide and complex transportation network and high-rise buildings. In other words, the information that can be obtained through GIS is vast, and these constraints are suitable for case studies compared with Nodeul Island.

Gwanghwamun is a representative cultural asset and tourist attraction in South Korea. Its most historic building, erected in 1395, attracts many tourists every year, and Gwanghwamun Square, which is approximately 730 m long, holds various events to entertain citizens. Therefore, Formula E could be held in Gwanghwamun, a potential future venue.

### 3.2. Data Pre-Processing

We incorporated the data in the QGIS environment on the NGII website for all information related to transportation networks, buildings, facilities, and topography within a radius of 2 km from Gwanghwamun (Figure 3). The constructed traffic-network data included 9210 road routes, 267.88 km of main roads, and 13,619 vertices intersecting between the main routes. This means that the average road network per unit $km^2$ was around 733; thus, the road network was very complex. In addition, we established and classified urban-infrastructure data corresponding to buildings and facilities based on the data-classification system described in Section 3.

Next, we extracted a dataset of usable road routes based on the factors set to a weight of 0 or 1. Table 5 shows the results of the data pre-processing. We aggregated the extracted data with 474 road routes, which had a cumulative length of 82.53 km; there were 1057 intersection vertices. We calculated the proportion of the data extracted through pre-processing as 5.15% of the road routes, 30.81% of the length of the road routes, and 7.76% of the intersection vertices. We extracted a higher percentage of the length of the road lines compared to the share of the extraction of the road lines and intersection vertices.

**Table 5.** Results of pre-processing.

| Data | Amount (Original) | Amount (Pre-Processing) | Ratio (%) |
|---|---|---|---|
| Road routes | 9210 | 474 | 5.15 |
| Cumulative length | 267.88 km | 82.53 km | 30.81 |
| Intersection vertices | 13,619 | 1057 | 7.76 |

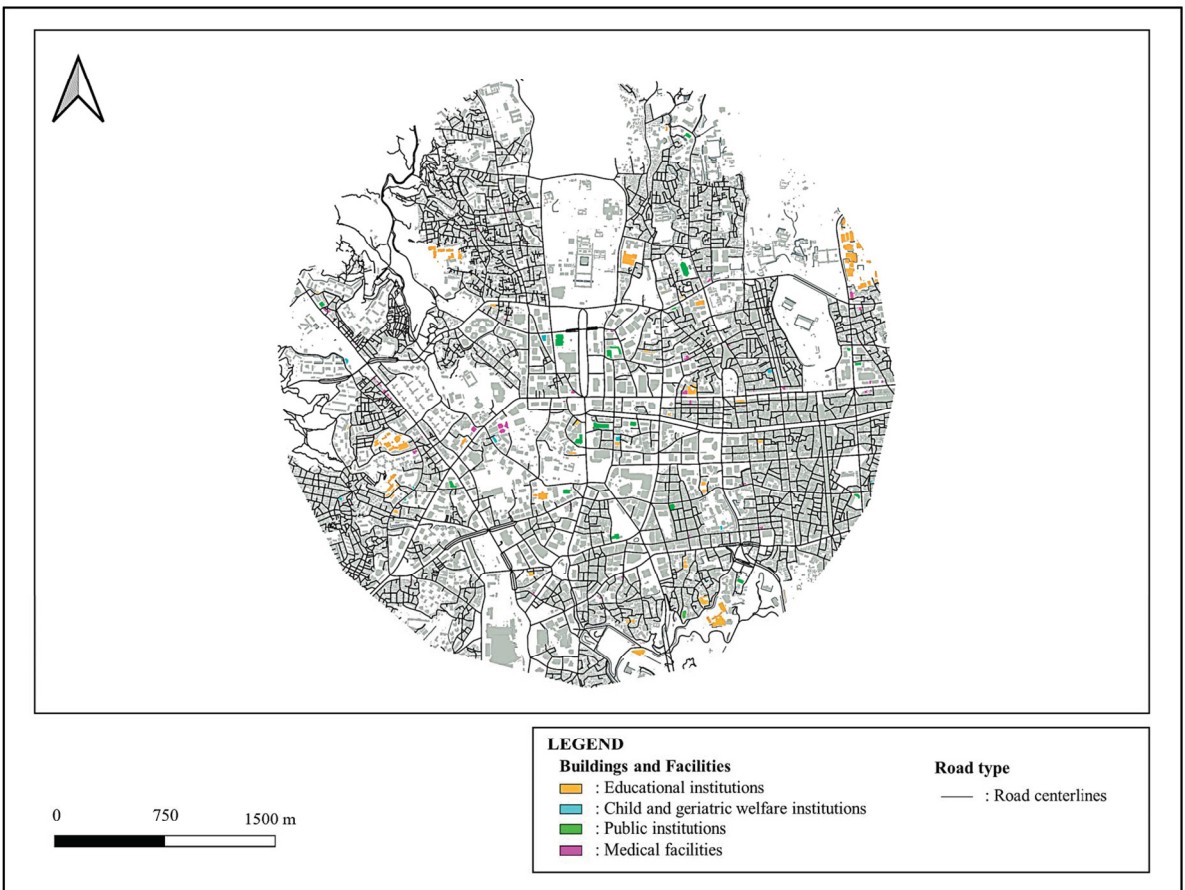

**Figure 3.** GIS data for Gwanghwamun area.

Subsequently, we processed the data that met the physical requirements of a road but which could not be used for a track. The removed data consisted of completely discontinuous roads, in which the starting and ending points were disconnected, and dead-end roads, where only one vertex was connected. We removed completely discontinuous roads entirely from the dataset, and we removed dead-end roads if the width of the road was less than the width required for a vehicle to turn (Figure 4).

### 3.3. Routing

Figures 5 and 6 display the optimal route derived through the GA process. The results are compared and displayed based on whether the environment variable was input. We returned the derived data in the form of closed curves and evaluated the suitability of the derived results based on the set parameters.

The conformity-assessment results are presented in Table 6. Result 1, which only considered route variables, produced suitable outcomes for track length and number of turns. However, the route passed public institutions a total of three times. In contrast, result 2, which considered environmental variables, passed facilities to be avoided only once. However, the number of turns fell below the standard value. This is a reasonable finding, which converges with the route factor of the city track required by the FIA and the environmental factor set to minimize the impact on the city. Furthermore, users can derive different paths by changing the factor parameters.

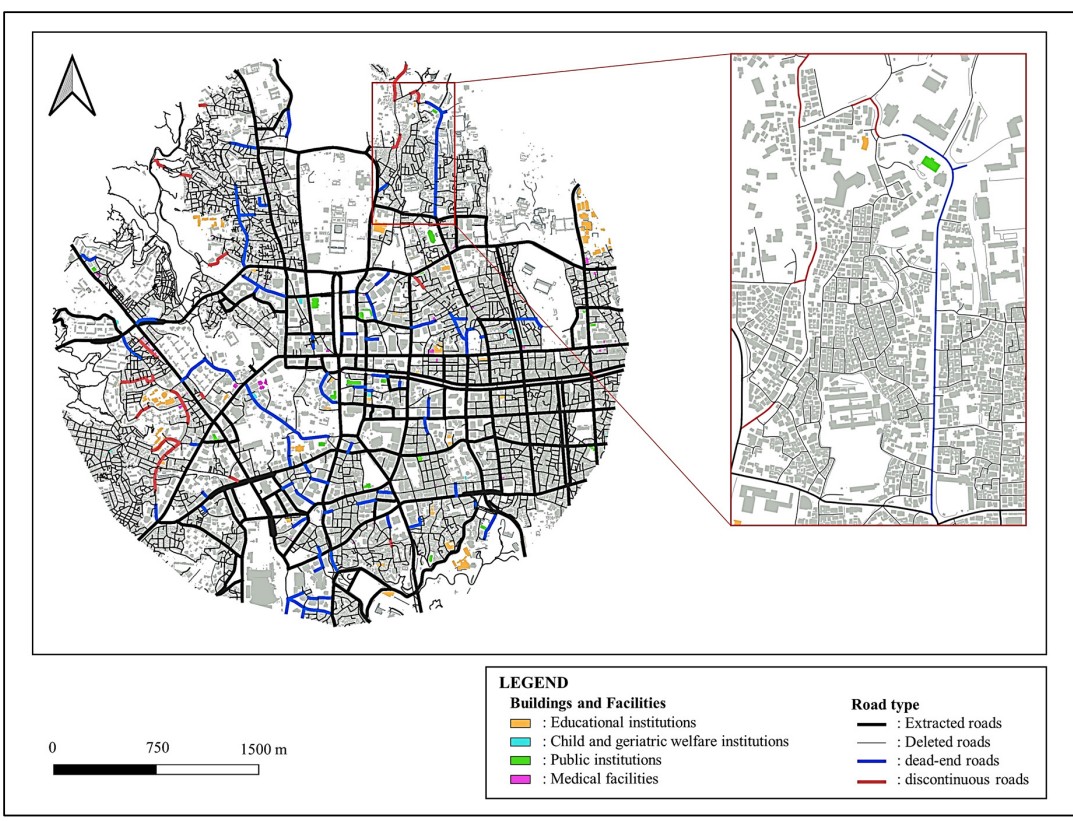

**Figure 4.** Data extracted through pre-processing.

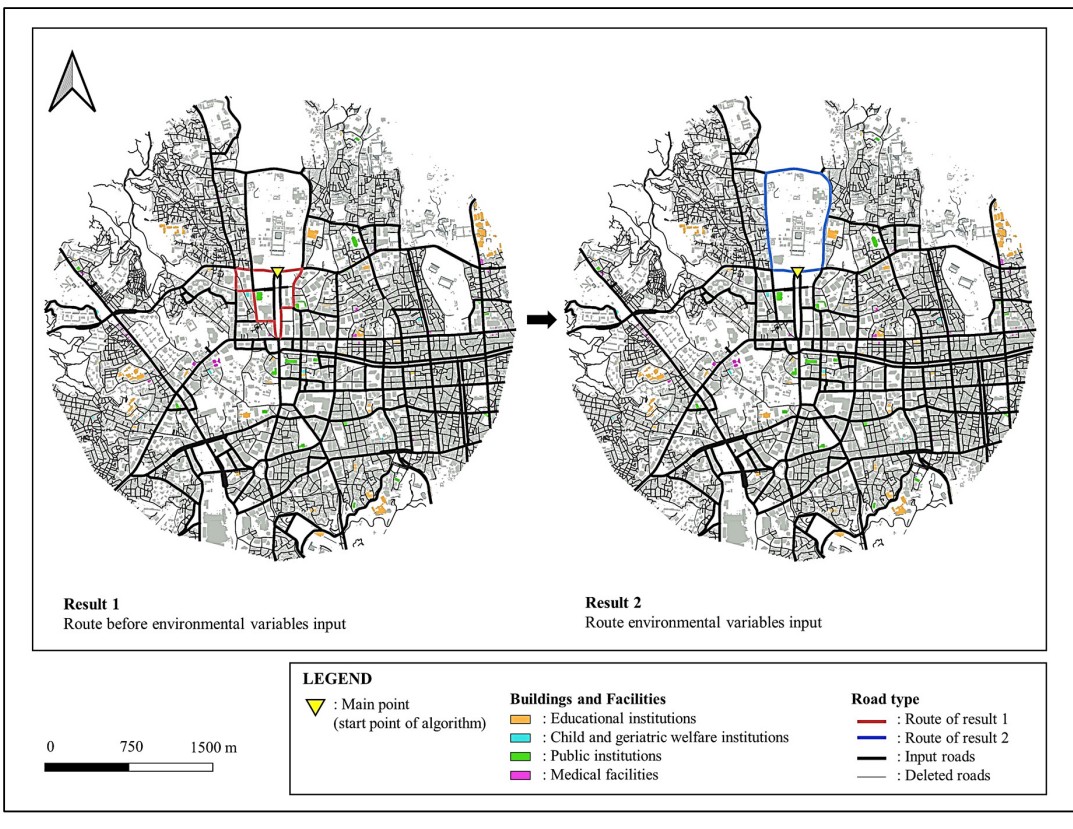

**Figure 5.** Optimal routes from the genetic algorithm—1.

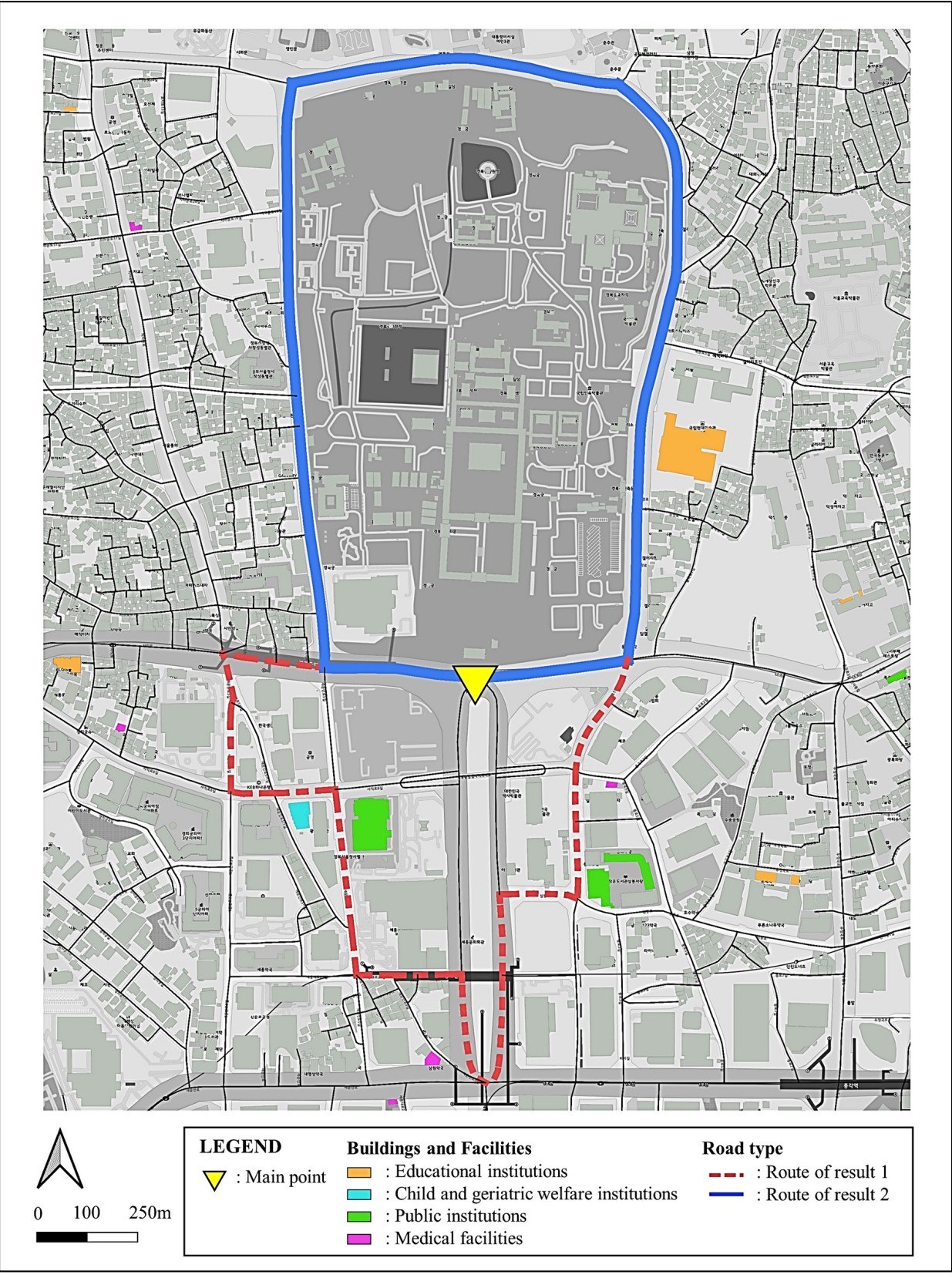

**Figure 6.** Optimal routes from the genetic algorithm—2.

**Table 6.** The conformity-assessment results of optimal routes.

| Category | Parameter | Value | Result 1 | Result 2 |
|---|---|---|---|---|
| **Level 01**:<br>Route variables—I | Width of the road (min.) | 12 M | 12.76 M | 14.14 M |
| | Pass through intersections | Y | Y | Y |
| | Pass through bridges | N | N | N |
| | Pass through tunnel | N | N | N |
| | Height obstacle (min.) | 4.5 | None | None |
| **Level 02**:<br>Route variables—II | Track length | 2.4 KM | 2.59 KM | 2.70 KM |
| | Tolerance range of track length | ±0.3 KM | +0.19 KM | +0.3 KM |
| | Number of turns (min.) | 14 turns | 14 | 11 |
| | Number of turns (max.) | 22 turns | | |
| | Radius of turning (max.) | 12 M | 12 M | 12 M |
| | Buffer interval length (min.) | 120 M | 120 M | 120 M |
| **Level 03**:<br>Environmental variables | Educational-institution avoidance | Y | Y | N (1 time) |
| | Child- and geriatric-welfare-institution avoidance | Y | Y | Y |
| | Public-institution avoidance | Y | N (3 time) | Y |
| | Medical-facility avoidance | Y | Y | Y |
| | Transportation-facility avoidance | Y | Y | Y |

### 3.4. GIS—BIM Integration

Figure 7 depicts the master BIM project model of the city, created within the BIM environment using the GIS-to-BIM process. The city model includes buildings, facilities, and roads with a radius of 2 km from Gwanghwamun, built in GIS in the form of a mass. In this process, we used the city model to evaluate the impact of the event on the city in a digital twin environment; the level of detail of the 3D shape is not important.

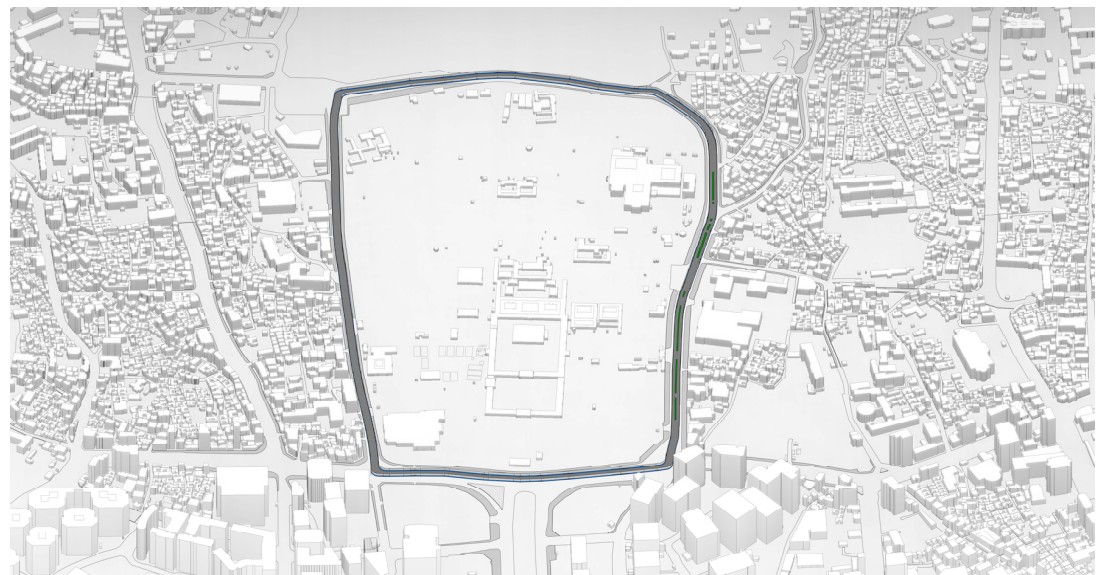

**Figure 7.** Master BIM project model of the city with optimal route.

The BIM data can be used to support decision-making. In this study, we examined the interference between the track route and urban infrastructure to support decision-making during the planning stage of the event (Figure 8). The interference review was simplified with the built-in inter-object-interference-review feature in Revit 2023. We reviewed a

total of eleven interferences; these refer to facilities that involved the demolition of urban infrastructures for the event. In addition, BIM data can be used to calculate demolition quantities and to visualize and manage projects.

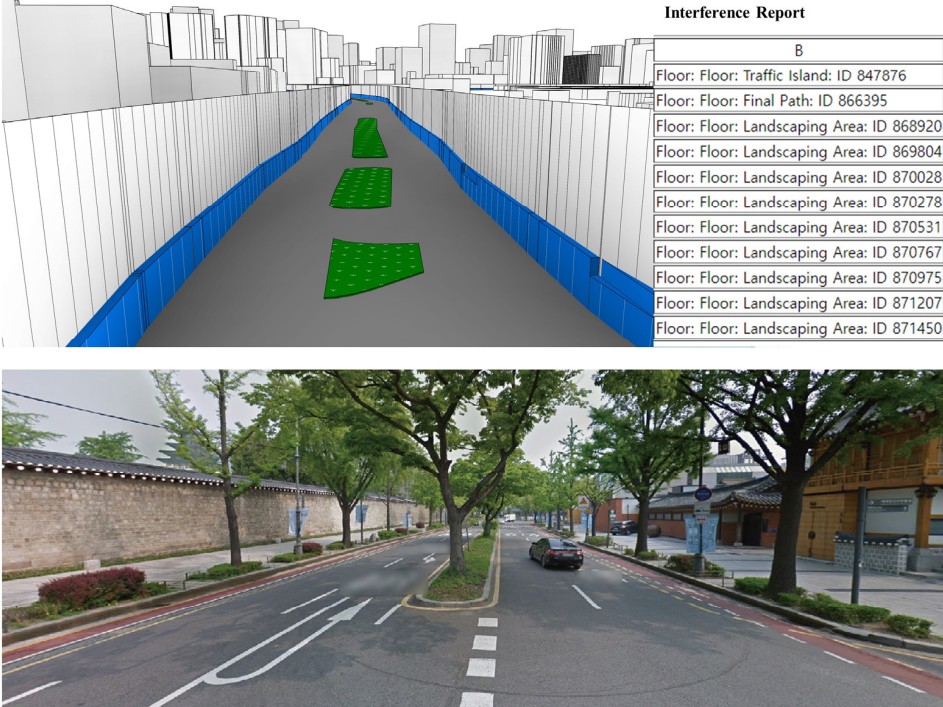

**Figure 8.** Example of interference between route and facilities.

## 4. Discussion

Mega events in cities can have positive social and cultural effects. Local governments and planners attempt to make public spaces more enjoyable through events held in cities. Events generate pleasure by facilitating direct and enjoyable experiences for citizens. However, it is inappropriate to degrade the public nature of a city for the sake of pleasure. Street-based sporting events that block roads for a period of time have a significant negative effects on host cities.

It is unrealistic to expect the complete elimination of the negative effects of mega events. However, events should be designed and managed such that public spaces can reach the common goal of inducing citizens' participation and protecting their public nature. Currently, many policymakers and event planners are attempting to reduce inconvenience for citizens, but many cities are continue to experience this inconvenience. Policymakers can improve the planning process (which has been limited to traditional event planning methods) with an approach based on massive data from cities, such as GIS.

In this study, we confirmed the applicability of the framework proposed in previous GIS-based routing studies for distance-based sports events. Furthermore, it was confirmed that by integrating the GIS-based routing process with the BIM environment, it is possible to effectively evaluate the impact on the downtown area. In addition, an algorithm that derives different routes according to the adjustment of environmental variables can search for suitable routes according to the needs of policymakers and event planners, and can quickly create various routes. Through the application of research on evaluation- and visualization-model creation through the integration of the current GIS and BIM, it was confirmed that this is an effective process for route evaluation. The technological-triad framework of GIS–routing–BIM proposed in this study can be applied to existing GIS-based routing research, and it is likely that an effective decision-making model can be implemented by integrating the research results in the BIM environment. Moreover, through the linkage of

BIM-based digital-twin technology, which is not covered in this study, it will be possible to implement a more powerful decision-making model.

The GIS- and algorithm-based event-path-selection process created using the proposed methodology can be utilized simply and effectively in the initial planning stage of an event. In addition, the technological-triad framework of GIS–routing–BIM proposed in this study can facilitate rapid and accurate decision making by policymakers. This can be used to address the negative effects of street-based mega-events by proactively identifying and preventing damage to public spaces. This methodology is meaningful because it presents an effective and rational digital method that breaks away from the existing planning process.

## 5. Conclusions

In this study, we developed a routing model based on GIS and a genetic algorithm to solve the limitations of the planning processes for traditional street-based sporting events. In addition, we applied the process of evaluating the impact on city infrastructure through data integration in the BIM environment.

We derived a GIS information system to search for routes for street-based sporting events targeting Seoul, and we defined the factors required for routing as route variables and environmental variables. We proposed and applied a routing process that obtains optimal results for a MOP by setting individual weights such that the defined factors could be applied as user parameters. It is possible for users to modify factors by adding variables or changing weights.

Moreover, we confirmed that data-based impact assessment is effective through the GIS-to-BIM process. The advantage of this technique is that it helps policymakers and event planners to proceed with easy-to-understand and prompt reviews of initial venue surveys. Therefore, this approach can be employed as a decision-making system before planning a full-scale event; it is objective, quick, and efficient compared with traditional feasibility studies for initial venues. In other words, in this study, an effective integration process was implemented through the GIS–routing–BIM technical-triad framework using the methodology presented in previous GIS–routing/GIS–BIM-dyad technological-combination studies.

This study suggests the possibility of implementing a more powerful and effective decision-making model when combining current GIS-based routing research with the BIM environment. In addition, the decision-making framework created through the BIM integration and routing model focused on distance-based sports events, which were not covered in previous GIS studies, can be applied not only to the Formula E World Championship but also to various distance-based sports events, such as marathons. Therefore, if the framework presented in this study is applied to future street-based events, it will support quick event-route searches and practical decision-making.

However, since this study was experimentally applied to a specific city, its effectiveness was not verified through a pilot application in actual street-based sports events. In addition, the fact that only one city, Gwanghwamun, was selected for the case verification is a limitation of this study. In addition, a further limitation of this study is that it did not explore the path variables required in various distance-based sports events, such as marathons, in selecting the Formula E Championship as a representative distance-based sports event. Moreover, the impact assessment through the GIS–BIM integration was focused on demolition and restoration work, which involve physical impact.

Therefore, in future research, it is necessary to collect the opinions of actual policy makers and project planners through pilot applications in real street-based sports events and, based on this, it is necessary to analyze the actual effects of this study. In addition, future research is needed to analyze the practical effects on decision making and the feasibility of conducting various impact assessments, such as a bypass-traffic-environment evaluation through a pedestrian simulation, and a traffic-congestion-impact evaluation through a traffic simulation, by linking the created BIM model with digital twin technology. Furthermore, follow-up research on optimal route-search and decision-making models considering various variables, such as pedestrians and vehicles, using the NSGA-III's MaOP

route-search methodology focusing on the complex simulation results of the digital-twin model is needed.

**Author Contributions:** Conceptualization, Y.-J.Y. and S.-Y.K.; methodology, Y.-J.Y. and S.-Y.K.; software, Y.-K.L.; validation, Y.-J.Y., S.-Y.K., Y.-J.Y., N.H. and J.-J.K.; formal analysis, Y.-J.Y. and S.-Y.K.; investigation, S.-Y.K. and Y.-K.L.; resources, Y.-J.Y., S.-Y.K., Y.-J.Y., N.H. and J.-J.K.; data curation, S.-Y.K.; writing—original draft preparation, Y.-J.Y. and S.-Y.K.; writing—review and editing, Y.-K.L., J.-H.K. and N.H.; visualization, Y.-K.L.; supervision, N.H.; project administration, Y.-J.Y., N.H., J.-H.K. and J.-J.K. All authors have read and agreed to the published version of the manuscript.

**Funding:** This work was supported by the National Research Foundation of Korea (NRF) grant funded by the Korea government (MSIT) (No. 2021R1F1A1052050).

**Institutional Review Board Statement:** Not applicable.

**Informed Consent Statement:** Not applicable.

**Data Availability Statement:** Not applicable.

**Conflicts of Interest:** The authors declare no conflict of interest.

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
