# Peer review of "GIS-Based Multi-Objective Routing Approach for Street-Based Sporting-Event Routing"

_applsci, doi:10.3390/app13148453_

Round 1
Reviewer 1 Report
The goal of the manuscript is to create a data- and algorithm-based routing mechanism for street-based events as well as to suggest a framework for assessing their influence on a particular host city. The authors developed an algorithm to look for the best event routes with the least negative impact on the city and built a database of host cities in a geographic information system (GIS). The information is also used to develop and evaluate building information modelling (BIM) models that evaluate urban impacts. The Seoul area of Gwanghwamun was chosen by the authors as a case study. To assess their impact on cities, they combined the BIM environment with GIS. They also created a routing model using a genetic algorithm and a geographic information system (GIS) to overcome the constraints of conventional street-based sporting events' planning procedures.
The research topic is quite fascinating. The work was well-designed, carried out, and presented, and
Corrections to be carried out
Section 1.2.1. The vehicle routing problem with GIS
Line no 97-141: (In literature on the vehicle ------- to minimize the impact on the city.) --- this part to be concise by highlighting the need of the present study
Reviewer 2 Report
The author demonstrates a high level of professionalism in the field of urban events planning based on GIS. However, there are still some concerning issues:
1.The core of this submitted research is the development of a new model; therefore, the core formula should be included in the paper to reflect this.
2.At the end of the abstract, it is inappropriate for the author to solely select the Gwanghwamun area in Seoul, South Korea, as the research case. It is necessary to provide additional research results, specifically the implementation effects of the case, to demonstrate the significance of the author's research work.
3.Lines 35-36: Placing this case here is inappropriate and requires modification or deletion.
None
Reviewer 3 Report
Overall, the proposed GIS-based multi-objective optimization approachis an interesting topic. However, the technical details and academic contributions of this study are not clearly expressed in the manuscript.
My specific comments and suggestions are as follows:
I recommend modifying the title of the paper to "GIS-Based Multi-Objective Routing Approach for Street-Based Sporting Event Routing" or a similar title. This study is focused on street-based sporting events in urban areas, not general urban events.
The Introduction section is too verbose and does not effectively summarize the research problem. The authors spend a lot of text introducing street-based sporting events in urban areas, but this is not necessary. Instead, the authors should provide a concise explanation of the characteristics of street-based sporting events and clearly define the importance of their research problem.
The authors used the NSGAII algorithm in their research, but there is no basic introduction on how this algorithm is used. For example, what are the optimization objective functions of NSGAII in their study? How are decision variables (chromosomes) encoded? What are the parameters of the NSGAII algorithm in the experiments?
The maps in Figures 2-4 are somewhat blurry.
I hope these comments are helpful in revising the manuscript for future submission.
English is difficult to understand
Reviewer 4 Report
Dear authors,
I reviewed the manuscript that developed a data- and algorithm-based routing process for street-based events and proposed a framework to evaluate their impact on a given host city. By integrating the data in a BIM framework, the study was able to evaluate the impact on city infrastructure. After carefully reading the manuscript, I feel that the idea is interesting. However, the manuscript does not meet the requirements for publication in the Journal of Applied Science.
Based on my understanding, the study lacked the ability to present a scientific discussion in a deep and comprehensive manner. This was to elaborate on the study's significance and explain how current problems have not been investigated by past studies. Furthermore, the manuscript failed to clearly generalize the study's findings beyond the case study.
For this reason, research lacks novelty to illustrate what research findings will add to existing knowledge.
Anyway, you can find my comments here:
Abstract:
Describe the research gap.
Mention the use of BIM in the abstract.
Instead of describing general findings, emphasize the research's novelty.
Introduction
The authors failed to provide evidence to support the current problem research.
This section is lacking in indicating how past studies support the research.
Literature Review
Generally, the review of the literature has not been completed, and the manuscript misses reviewing the theoretical background. I see there is no discussion regarding the relevant theories, models, and concepts in this field of knowledge.
It is important for the authors to use recent publications when conducting this section.
Discussion:
Through this part, authors must discuss research results and connect with past studies.
Conclusion
Here, the author should describe the theoretical and practical implications, research contributions, and limitations of the current work. In addition, the author should make suggestions for future research.
As I see, the authors reviewed the manuscript in this part, which needs revision.
Round 2
Reviewer 3 Report
The author has made substantial revisions to the paper and has addressed all my concerns
Author Response
Dear Reviewer 2,
We, all of the authors, sincerely appreciate your taking time to review for our submission
Thank you for approving this paper.
Yours sincerely,
Namhyuk Ham
Reviewer 4 Report
Dear authors,
Thank you for your efforts to improve the manuscript. I’ve read the manuscript and the author's reply, but I still see major problems with the manuscript.
Based on the first round of comments, and on my own reading, I am afraid I cannot accept the manuscript as it stands. You claimed that the comments had been addressed, but the major issues are seen in the manuscript.
However, you can find my comments here:
Based on my understanding, the study still lacked the ability to illustrate a scientific discussion in a deep and comprehensive manner. This was to elaborate on the study's significance and explain how current problems have not been investigated by past studies.
In addition, the manuscript has failed to generalize the study's findings beyond the case study. For this reason, I feel the lack of research novelty is serious and the authors are unable to illustrate what research findings will add to existing knowledge.
Abstract:
I asked you to describe the research gap, but I didn’t find it. You need to highlight the changes with a color font.
I asked you to emphasize the research's novelty instead of describing general findings, but I didn’t see it.
Introduction
I indicated that the Introduction lacked evidence to support the current problem research. Based on the revised manuscript, it still lacks evidence.
Literature Review and hypothesis development
Based on the first-round comments, it was highlighted that the manuscript missed reviewing the theoretical background. I see there is not a discussion of relevant theories or models. The authors just mentioned the NSGA-III approach which is not enough.
Discussion:
Researchers were asked in this part to discuss research results and connect them with previous studies, but the revised manuscript failed to take that into account.
Conclusion
Here, the author didn’t address the required information based on my understanding. There is no statement about the theoretical and practical implications, research contributions, and limitations of the current work. In addition, the author failed to make suggestions for future research.
Author Response
첨부 파일을 참조하십시오.

Round 3
Reviewer 4 Report
Dear authors,
After reading the revised manuscript, I found that most of the comments were appropriately explained and justified by the authors. Based on my understanding, the manuscript met the journal’s requirements for publication.
So, I don't have more questions.